# Impact of clinical supervision on healthcare organisational outcomes: A mixed methods systematic review

Priya Martin[1,2]*, Lucylynn Lizarondo[3], Saravana Kumar[4], David Snowdon[5,6]

**1** Senior Research Fellow, Rural Clinical School, Faculty of Medicine, The University of Queensland, Toowoomba, QLD, Australia, **2** Advanced Clinical Educator Interprofessional, Advance Queensland Industry Research Fellow, Cunningham Centre, Darling Downs Health, Toowoomba, QLD, Australia, **3** Research Fellow: Implementation Science, Joanna Briggs Institute, The University of Adelaide, Adelaide, SA, Australia, **4** Allied Health and Human Performance, University of South Australia, Adelaide, SA, Australia, **5** Research Fellow, Peninsula Clinical School, Central Clinical School, Monash University, Melbourne, VIC, Australia, **6** Allied Health Research Lead, Academic Unit, Peninsula Health, Melbourne, VIC, Australia

\* Priya.Martin@uq.edu.au

## Abstract

### Objective

To review the impact of clinical supervision of post-registration/qualification healthcare professionals on healthcare organisational outcomes.

### Background

Clinical supervision is a professional support mechanism that benefits patients, healthcare professionals and healthcare organisations. Whilst evidence is growing on the impact of clinical supervision on patient and healthcare professional outcomes, the evidence base for the impact of clinical supervision on organisational outcomes remains weak.

### Methods

This review used a convergent segregated approach to synthesise and integrate quantitative and qualitative research findings, as per the Joanna Briggs Institute's recommendations for mixed methods systematic reviews. Databases searched included CINAHL, Embase, PubMed, PschINFO, and Scopus. Whilst a narrative synthesis was performed to present the findings of the quantitative and qualitative studies, the evidence from both quantitative and qualitative studies was subsequently integrated for a combined presentation. The review followed the guidelines of the Preferred Reporting Items for Systematic Reviews and Meta-Analyses.

### Results

Thirty-two studies including 27 quantitative, two qualitative and three mixed methods studies, were included in the review. The results of the quantitative analysis showed that effective clinical supervision was associated with lower burnout and greater staff retention, and effective supervisor was associated with lower burnout and greater job satisfaction.

**Data Availability Statement:** All relevant data are within the manuscript and its Supporting Information files.

**Funding:** The authors received no specific funding for this work

**Competing interests:** The authors have declared that no competing interests exist.

Qualitative findings showed that healthcare professionals believed that adequate clinical supervision could mitigate the risk of burnout, facilitate staff retention, and improve the work environment, while inadequate clinical supervision can lead to stress and burnout. The evidence from quantitative and qualitative studies were complementary of each other.

## Conclusion

Clinical supervision can have a variable effect on healthcare organisational outcomes. The direction of this effect appears to be influenced by the effectiveness of both the clinical supervision provided and that of the clinical supervisor. This highlights the need for organisations to invest in high quality supervision practices if maximal gains from clinical supervision are to be attained.

## Introduction

Clinical supervision is widely practiced in health and social care professions across the globe owing to its beneficial effects to patients, health professionals and organisations [1, 2]. Operationally, clinical supervision, for post-qualification health professionals, is viewed as a process that provides quarantined time and an opportunity to further develop the supervisee's skills and knowledge, within the context of an ongoing professional relationship, usually with an experienced practitioner (one-to-one supervision), or with peers (peer group supervision). The aim of clinical supervision is for the supervisee to engage in guided reflection on current practice in ways designed to develop and enhance that practice in the future [1, 2]. This type of supervision involves reflective thinking, and discussion regarding professional development issues, caseload, clinical issues, and staff interpersonal issues. Issues in clinical supervision definition and terminologies are widely prevalent [2]. In this review, the following definition of clinical supervision has been adopted:

> "*The formal provision, by approved supervisors, of relationship-based education and training that is work-focused, and which manages, supports, develops and evaluates the work of colleague/s*" [1].

Whilst efforts are growing to strengthen the evidence for clinical supervision, there is also criticism about a vast majority of evidence on supervision, as being *proof by association* or tentative [3]. While there is a growing evidence base for the impact of clinical supervision on patient outcomes such as reduced risk of mortality, reduced risk of complications and more effective care [4–7], and health professional outcomes such as being better supported in their roles [8], there remains a need to systematically review the evidence for the impact of clinical supervision of post-qualification health professionals, on organisational outcomes, to further strengthen the evidence base on clinical supervision.

Determining the impact of clinical supervision on healthcare organisations, however, is difficult given the challenges in defining organisational outcomes and the overlapping nature of patient, health professional and organisational outcomes. For example, improved patient outcomes (e.g. improved morbidity and mortality) can satisfy multiple targets for healthcare organisations, as can health professional outcomes (e.g. reduction in stress and burnout), which can reduce staff sick leave, a usual key performance indicator for organisations. In determining the organisational outcomes of interest for this review, we undertook a scan of the broader

literature. A recent systematic review of leadership styles and outcome patterns for the nursing workforce and work environment, grouped the outcomes into six categories: staff satisfaction and job factors, staff relationships with work, staff health and wellbeing, relations among staff, organisational environment factors and productivity and effectiveness [9]. Another systematic review on the relationship between governance mechanisms in healthcare and health workforce outcomes considered staff turnover and job satisfaction [10]. Other organisational outcomes cited in the clinical supervision literature include improved teamwork [11] and job satisfaction [12]. In considering all this, organisational outcomes in the current review will reflect the well-being of health professionals resulting from clinical supervision, that lead to better outcomes for the organisations such as recruitment and retention, intent-to-stay, intent-to-leave, job satisfaction and quality of work life, burnout and absenteeism. Furthermore, despite the benefits of supervision, to date, no review has explored health professionals' perspectives of, and the impact from, clinical supervision on organisational outcomes.

Therefore, as means of addressing these knowledge gaps, using a mixed methods design, this review aims to answer the following research questions:

1. What are the effects of clinical supervision of healthcare professionals on organisational outcomes?

2. What are healthcare professionals' experiences, views, and opinions regarding clinical supervision as it relates to organisational processes and outcomes?

3. What can be inferred from the qualitative synthesis of healthcare professionals' experiences/ views that can explain the effects of clinical supervision or inform its appropriateness and acceptability for health professionals?

## Methods

This systematic review was conducted using Joanna Briggs Institute (JBI) methodology for mixed methods systematic review, specifically the convergent segregated approach to synthesis and integration [13]. The review followed the Preferred Reporting Items for Systematic Reviews and Meta-Analyses (PRISMA) guideline [14] and was based on an a-priori published protocol [15].

### Eligibility criteria

The review protocol indicated the inclusion of studies that focused on one-to-one clinical supervision rather than group supervision. However, during the screening of studies, it became apparent that there was a prevalence of studies that investigated both one-to-one and group supervision (which was facilitated by a supervisor, as opposed to peer supervision), and studies that did not specify the type of clinical supervision investigated. Given this challenge, and to reflect the reality of healthcare organisations utilising both these types of supervision regularly, the review team agreed to include any study on clinical supervision, regardless of the type (i.e. one-to-one or group). To be eligible, studies had to meet the following criteria: (1) investigated clinical supervision of qualified or registered health professionals (i.e. clinical supervision of post-qualified health professionals, where they engage in one-to-one or group supervision sessions that happen over a period of time); (2) used qualitative, quantitative or mixed-methods study design; (3) if a quantitative study, examined the effects of clinical supervision on organisational outcomes, such as staff retention and recruitment, intent to stay, intent to leave, job satisfaction and quality of work life, burnout, and absenteeism; (4) if a

qualitative study, explored health professionals' experiences, views, or opinions regarding clinical supervision as they relate to organisational outcomes.

## Search strategy

As means of avoiding publication and location bias, the search strategy was developed to identify black (commercially published) and grey literature. Search terms were identified based on the key concepts relating to the intervention/phenomenon of interest, i.e. clinical supervision and outcomes of interest, i.e. organisational outcomes.

An initial limited search of PubMed and CINAHL was undertaken followed by analysis of text words contained in the title and abstract and the index terms used to describe the articles. The search strategy, including all identified keywords and index terms, was then adapted for each database. The search for published studies was performed from the date of inception until May 2020 in the following databases: CINAHL, Embase, PubMed, PsycINFO, and Scopus. These databases were chosen as they commonly include literature from health disciplines, a combination of discipline specific (e.g. CINAHL includes nursing and allied health literature) and multi-disciplinary (e.g. Scopus) and are routinely used in systematic reviews. The search for grey literature was undertaken in ProQuest Dissertations and Theses, Google Scholar and WorldWideScience.org. Reference lists of relevant studies were reviewed to identify additional publications. The search strategy for each database is shown in S1 Appendix.

## Study selection

Following the search, all identified citations were collated and uploaded into EndNote X8.2 (Clarivate Analytics, PA, USA) [16] and duplicates removed. Two reviewers independently screened the titles and abstracts (LL and DS) against the inclusion criteria for the review. Potentially relevant articles were retrieved in full and assessed independently for eligibility by two other reviewers (PM and SK). Disagreements were resolved through discussion and consensus. Studies that did not meet the inclusion criteria were excluded and reasons for their exclusion are provided in S2 Appendix. Abstracts and full text articles did not require translation to another language to determine their eligibility. All full text articles reviewed contained sufficient information to determine their eligibility without the need for further clarification from authors. The PRISMA flow diagram of included studies is available in Fig 1.

## Quality assessment

All eligible studies were assessed for methodological quality by two independent reviewers (PM and DS for quantitative studies; PM and LL for qualitative studies using the relevant JBI critical appraisal tools [17]. These tools were chosen as they assist in assessing the trustworthiness, relevance and results of published studies and are widely used. Any disagreements that arose between the reviewers were resolved through discussion. All studies, regardless of the results of their methodological quality, underwent data extraction and synthesis.

## Data collection

For the quantitative component, data were extracted from quantitative and mixed methods studies (quantitative component only) and included specific details about the supervisee and supervisor characteristics (sample size, profession), characteristics of the supervision (type, frequency, duration), study design, setting, clinical supervision characteristics, outcomes measured, and results related to the organisational outcomes. For the qualitative component, data were extracted from qualitative and mixed methods studies (qualitative component only) and

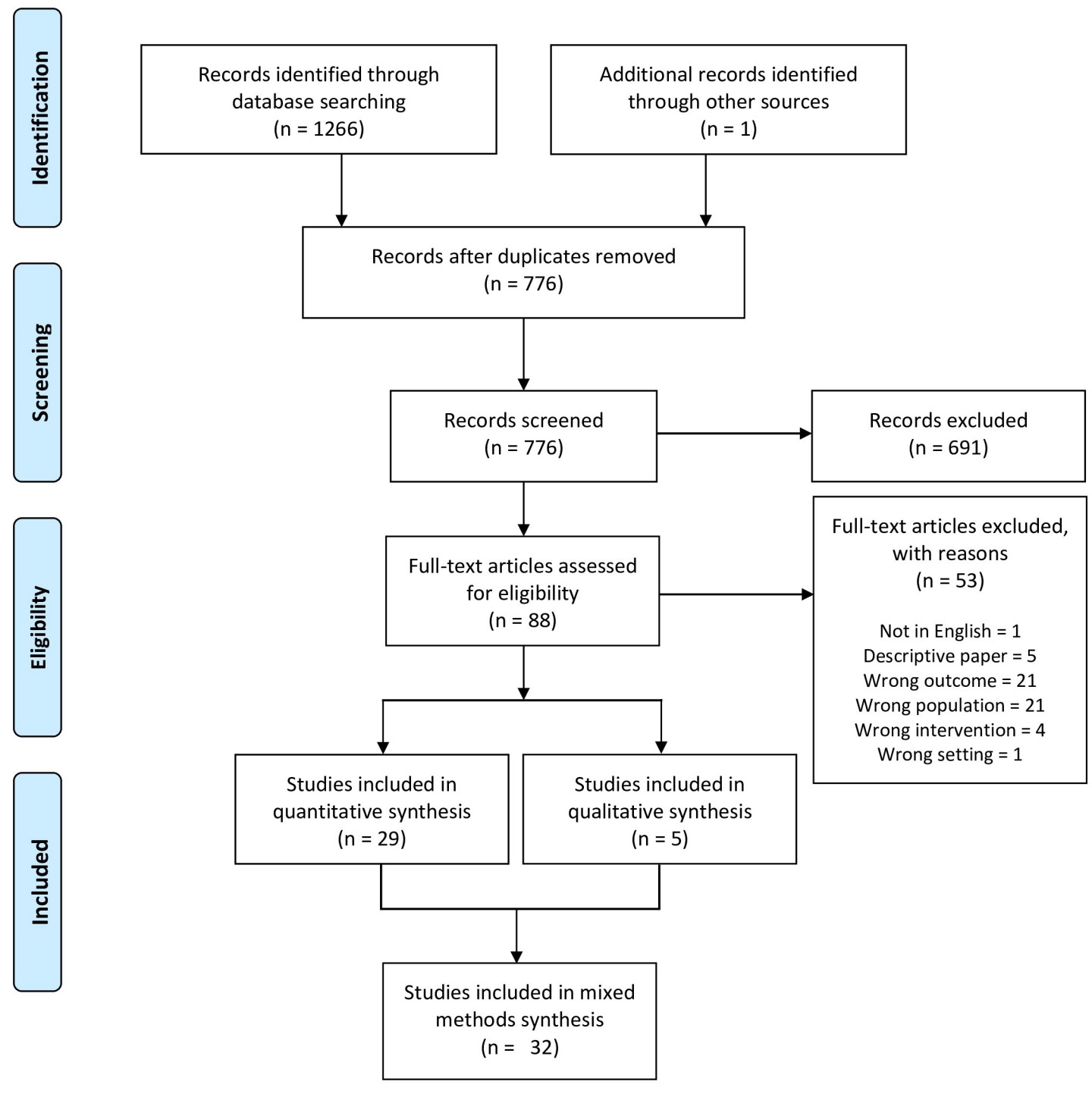

**Fig 1. Flow diagram of included studies.**

included specific details about the supervisee and supervisor characteristics (sample size, profession, work experience), study design and methods, setting, and findings which included participants' experiences of clinical supervision as they relate to organisational outcomes. Findings extracted from individual studies consisted of themes or subthemes reported by the

authors. These findings were accompanied by a direct quotation representing a participant's voice (i.e. illustration). Findings were also assigned one of three levels of credibility according to the following criteria: (1) unequivocal: findings accompanied by an illustration that is beyond reasonable doubt and therefore not open to challenge, (2) credible: findings accompanied by an illustration lacking clear association with it and therefore open to challenge, and (3) not supported: findings are not supported by data. The review team discussed the data extraction process, established standards and consistencies on how this should occur, and those with quantitative expertise (DS) and qualitative expertise (LL) lead the extraction process, with the primary reviewer (PM) acting as the additional reviewer for validation purposes.

## Data synthesis and integration

A convergent segregated approach to synthesis and integration was applied [13]. This involved an initial independent synthesis of the quantitative studies and qualitative studies followed by the integration of findings from such syntheses using configurative analysis.

Quantitative data were analysed descriptively; meta-analyses were deemed not appropriate due to heterogeneity between studies in terms of clinical supervision interventions and participants. Odds ratios (OR) of dichotomous events and standardised mean differences (SMD) for continuous measures were calculated. For experimental studies OR were converted to SMD using an online calculator [18], to assist with interpretation of effect size. For observational analytical studies the correlational coefficient ($r$) was calculated in addition to OR and SMD. Effect size was determined using the following reference values for SMD: small 0.2, medium 0.5, large 0.8 [19]; OR: small 1.68, medium 3.47, large 6.71 [20]; and $r$: small 0.1, moderate 0.3, large 0.5 [19].

Qualitative synthesis was conducted using the meta-aggregative approach [21]. Meta-aggregation is aligned with the philosophy of pragmatism, focusing on the practicality and usability of the synthesised findings and generation of statements that are useful for informing actions in clinical practice [21]. This involved assembling and aggregating the extracted findings from individual studies, based on similarity in meaning, to generate a set of statements (i.e. categories) that represented that aggregation. These categories were then subjected to meta-synthesis to produce a set of synthesised findings. The development of categories and synthesised findings was conducted via a consensus process between the reviewers (LL and PM).

The findings of each single method synthesis were juxtaposed and examined for convergence/divergence and complementarity. To explore the relationship across individual syntheses, the findings were reviewed to determine whether they were supportive or contradictory and identify which aspects of the quantitative evidence were not explored in the qualitative studies and vice-versa. The clinical supervision interventions which had been investigated in the quantitative studies were further analysed in line with the experiences of participants in the qualitative studies to explain the impact of clinical supervision on the different organisational outcomes. However, due to the heterogeneity of the quantitative studies and the lack of well-conducted trials, and the limited qualitative studies, no clear cause and effect relationships can be determined, nor in-depth analysis can be made to explain the impact of clinical supervision.

## Results

The database search yielded 1266 records. Eighty-five articles were retrieved for full text review following application of the eligibility criteria to title and abstract. Thirty-four fulfilled the inclusion criteria when applied to full texts. Three of these articles were duplicate publications,

resulting in a yield of 31 studies. One article was identified through pearling of references in the included studies; hence the final yield was 32 studies (Fig 1).

## Study characteristics

Twenty-seven quantitative [22–48], two qualitative [49, 50] and three mixed methods studies [51–53] were included in the review. Fifteen studies used a randomised controlled (n = 1) [22] or quasi-experimental design (n = 14) [23–35, 51] to establish the effect of clinical supervision on organisational outcomes. Eight studies investigated the association between effectiveness of clinical supervision and organisational outcomes [37–43, 52]. Eight studies investigated the association between the effectiveness of the supervisor and organisational outcomes [32, 36, 39, 41, 43, 45, 46, 48]. Two studies used a cross sectional survey study design to measure perceptions of effect of clinical supervision on organisational outcomes [44, 47]. Four studies [49–52] used a qualitative descriptive design, with either individual [49–51] or focus group [52], semi structured interviews as the method of data collection. The qualitative component of one study [53] applied the grounded theory methodology, using a qualitative questionnaire for data collection. Ten studies were published in the 1990s [23, 25, 26, 30, 34, 45–49], six studies were published in the 2000s [24, 28, 35, 39, 41, 43], and 16 studies were published in the 2010s [22, 27, 29, 31, 32, 33, 36–38, 40, 42, 44, 50–53] with seven of these published in the last 5 years [27, 37, 40, 50–53].

Studies were conducted in hospital (n = 15) [22–26, 29–31, 34, 37, 40, 41, 44, 47, 53], community healthcare settings (n = 6) [32, 36, 38, 39, 48, 52] and a combination of hospital and community healthcare settings (n = 11) [27, 28, 33, 35, 42, 43, 45, 46, 49–51]. Most studies were conducted in the mental health setting (n = 15) [25–28, 30, 34, 37, 39, 42–44, 46–48, 53]. Health professionals who received clinical supervision included nursing (n = 23) [22–26, 30–35, 37, 39–42, 44, 46–49, 51, 53], social work/psychology/counselling professionals (n = 10) [27, 32, 36, 38, 43, 45, 46, 48, 52], other allied health professionals (n = 4) [28, 29, 50, 52] and medical professionals (n = 3) [22, 33, 48]. Seven studies were conducted in Sweden, [23, 25, 26, 28, 30, 34, 47] seven in Australia [32, 38, 40, 43, 50–52], seven in the United Kingdom [22, 35, 37, 39, 44, 49, 53], four in the United States of America [27, 45, 46, 48], two in Finland [31, 41] and one each in Norway [24], Israel [36], Africa [33], Denmark [42] and Italy [29]. Eight studies investigated only group supervision [23–26, 30, 31, 42, 47] four studies investigated only individual (one-to-one) supervision [22, 27, 38, 50], 12 studies investigated both group and individual supervision [29, 32, 35, 39, 41, 43–46, 49, 51, 52] and eight studies did not state whether the supervision they investigated was group or individual [28, 33, 34, 36, 37, 40, 48, 53]. The frequency and duration of supervision sessions were variable between studies, ranging from weekly to every three months, and 30 to 480 minutes. Frequency and duration of supervision were not reported in 16 [22, 28, 29, 33–37, 40, 45, 48–53] and 18 studies [24, 28, 29, 33–38, 40, 44, 45, 48–53], respectively.

Five studies (two qualitative [49, 50] and three mixed methods studies [51–53]) explored the clinical supervision experiences of healthcare professionals including its impact on clinical practice. Fifteen studies investigated the effect of supervision on burnout [22, 25, 27–32, 35–37, 39, 41, 42, 48, 52], 9 studies on other measures of well-being [22, 24–26, 30–32, 42, 44], 13 studies on job satisfaction [25–28, 30, 32, 33, 41–43, 45, 46, 51], 9 studies on the work environment [23–26, 31, 34, 35, 38, 47], and 3 studies on job retention [32, 33, 40]. There was a large diversity of outcome measures used with only four measures used in more than one study; the Maslach Burnout Inventory was used in 13 studies [25, 27, 29–32, 35, 37, 39, 41, 42, 48, 52], and the Creative Climate Questionnaire [25, 26], Tedium Measure [25, 30] and Satisfaction with Nursing Care questionnaire [25, 30] each used in two studies. Study characteristics can be found in Table 1.

**Table 1. Study characteristics.**

| Study | Design | Setting | Participants | | Supervision | | | | Outcomes (Quantitative) OR Interview questions (Qualitative) |
|---|---|---|---|---|---|---|---|---|---|
| | | (country) | Supervisee | Supervisor | Type | Frequency | Duration | | |
| | | | **Profession** | **Profession** | | | | | |
| | | | **Work Experience,** *mean* | | | | | | |
| | | | **n** | | | | | | |
| Begat 1997 | Quantitative Quasi-experimental pre/post | Acute hospital medical wards (Sweden) | Nursing / 11 to 18 years / n = 34 | Nursing | Group | Weekly—Fortnightly | 90 minutes | | **Work environment**[a] |
| Begat 2005 | Quantitative Quasi-experimental cross sectional | Acute hospital medical wards (Norway) | Nursing / 9 years / n = 71 | N/S | Group | Fortnightly | N/S | | **Well-Being**[a] / **Work Environment** / WEQ |
| Ben-Porat 2011 | Quantitative Cross sectional | Domestic violence and women's shelters (Israel) | Social Work / 11 years / n = 143 | N/S | N/S | N/S | N/S | | **Burnout** / Burnout Questionnaire |
| Berg 1994 | Quantitative Quasi-experimental pre/post | Psychogeriatric hospital (Sweden) | Nursing / 11 years / n = 39 | Nursing | Group | Fortnightly–every third week | 120 minutes | | **Burnout** / MBI / **Job Satisfaction** / Satisfaction with Nursing Care / **Well-being** / Tedium Measure / **Work Environment** CCQ |
| Berg 1999 | Quantitative Quasi-experimental pre/post | Psychiatric hospital (Sweden) | Nursing / 14 years / n = 22 | Nursing | Group | Fortnightly | 180 minutes | | **Job Satisfaction** / SNCW / **Well-being** / SOC / WRSI / **Work Environment** / CCQ |
| Berry 2019 | Quantitative Cross sectional | Psychiatric hospital (UK) | Nursing / N/S / n = 137 | N/S | N/S | N/S | N/S | | **Burnout** MBI |
| Best 2014 | Quantitative Cross sectional | Alcohol and drug community service (Australia) | Social Work/ Psychology/ Counselling / 56% > 10 years / n = 43 | N/S | Individual | Fortnightly–monthly | N/S | | **Work Environment** / Organizational Readiness for Change Assessment |
| Cooper-Nurse 2018 | Quantitative Quasi-experimental cross sectional | Mental health settings (USA) | Social Work/ Psychology/ Counselling / N/S / n = 60 | N/S | Individual face-to-face +/- over phone/online | 55% less than once per week | 82% >30 minutes | | **Burnout** / MBI / **Job Satisfaction** / AJDI |

*(Continued)*

**Table 1.** (Continued)

| Study | Design | Setting | Participants | | Supervision | | | Outcomes (Quantitative) OR Interview questions (Qualitative) |
|---|---|---|---|---|---|---|---|---|
| | | (country) | Supervisee | Supervisor | Type | Frequency | Duration | |
| | | | Profession | Profession | | | | |
| | | | Work Experience, *mean* | | | | | |
| | | | n | | | | | |
| Ducat 2016 | Qualitative Qualitative descriptive | Rural and regional areas (Australia) | Social work/ Nutrition/Dietetics/ Occupational Therapy/ Physiotherapy/ Speech pathology/ Medical radiation/ Psychology | N/S | Individual | N/S | N/S | **Interview question** |
| | | | N/S | | | | | What effect has CS had on your practice (if any)? |
| | | | n = 42 | | | | | |
| Edwards 2006 | Quantitative Cross sectional | Community mental health (UK) | Nursing | N/S | Individual, group or combination | 57% monthly | 32% >60 minutes | **Burnout** |
| | | | 52% <5 years | | | | | MBI |
| | | | n = 260 | | | | | |
| Eklund 2000 | Quantitative Quasi-experimental cross sectional | Acute and community psychiatric care (Sweden) | Occupational Therapy | Occupational Therapy/Social Work/ Psychology Nursing/ Medical | N/S | N/S | N/S | **Job Satisfaction** |
| | | | N/S | | | | | Job Satisfaction Questionnaire |
| | | | n = 291 | | | | | |
| Fischer 2013 | Quantitative Quasi-experimental cross sectional | Acute Hospital (Italy) | Physiotherapy | N/S | Individual or group | N/S | N/S | **Burnout** |
| | | | 13 years | | | | | MBI |
| | | | n = 132 | | | | | |
| Gonge 2011 | Quantitative Cross sectional | Psychiatric hospital wards and community mental health centres (Denmark) | Nursing | Psychiatry/ Psychology | Group | Every two months | 90 minutes | **Burnout** |
| | | | | | | | | MBI |
| | | | | | | | | **Job Satisfaction** |
| | | | N/S | | | | | CPQ |
| | | | | | | | | **Well-being** |
| | | | | | | | | CPQ |
| | | | n = 145 | | | | | SF-36 |
| Hallberg 1994 | Quantitative Quasi-experimental pre/post | Paediatric psychiatric ward (Sweden) | Nursing | Nursing | Group | Every third week | 120 minutes | **Burnout** |
| | | | | | | | | MBI |
| | | | | | | | | **Job Satisfaction** |
| | | | 15 years | | | | | Satisfaction with Nursing Care |
| | | | | | | | | **Well-being** |
| | | | n = 11 | | | | | Tedium Measure |
| Hussein 2019 | Quantitative Cross sectional | Acute hospital (Australia) | Nursing | N/S | N/S | N/S | N/S | **Job Retention** |
| | | | 1 year | | | | | Modified Nurse Retention Index |
| | | | n = 87 | | | | | |

(*Continued*)

**Table 1.** (Continued)

| Study | Design | Setting | Participants | | Supervision | | | Outcomes (Quantitative) OR Interview questions (Qualitative) |
|---|---|---|---|---|---|---|---|---|
| | | (country) | Supervisee<br><br>**Profession**<br><br>**Work Experience,** *mean*<br><br>**n** | Supervisor<br><br>**Profession** | Type | Frequency | Duration | |
| Hyrkäs 2005 | Quantitative Cross sectional | Acute hospitals (Finland) | Nursing<br>57% > 10 years<br>n = 569 | Nursing/ Psychology | Individual or group | 67% every three weeks or monthly | 34% 60 minutes duration | **Burnout**<br>MBI<br>**Job Satisfaction**<br>Minnesota Job Satisfaction Scale |
| Kavanagh 2003 | Quantitative Cross sectional | Hospital and community mental health settings (Australia) | Social Work/ Psychology/ Occupational Therapy/ Speech Therapy<br>8 years<br>n = 199 | N/S | Individual, group or combination | Monthly | 120 minutes | **Job Satisfaction**<br><br>Hoppock Job Satisfaction Measure |
| Koivu 2012 | Quantitative Quasi-experimental cross sectional | Acute hospital medical and surgical wards (Finland) | Nursing<br>15 to 17 years<br>n = 304 | N/S | Group | Every 3 or 4 weeks | 90 minutes | **Burnout**<br>MBI-GS<br>**Well-being**<br>GHQ-12<br>**Work Environment**<br>QPSNordic |
| Livini 2012 | Quantitative Quasi-experimental pre/post | Drug and alcohol service (Australia) | Nursing/ Psychology/Social Work/Counselling<br>N/S<br>n = 42 | Nursing/ Psychology | Individual, group or combination | 2 to 8 sessions over 6 months | 70 to 480 minutes | **Burnout**<br>MBI<br>**Job Satisfaction**<br>IJSS<br>**Well-being**<br>Scales of psychological well-being |
| Long 2014 | Quantitative Cross sectional | Mental Health Hospital (UK) | Nursing<br>28% > 7 years<br>n = 128 | N/S | Individual, group or combination | 23% monthly | N/S | **Well-being**<br>BCS |
| Love 2017 | Quantitative Quasi-experimental cross sectional Qualitative Qualitative descriptive | Hospital and community maternity services (Australia) | Nursing<br>17 years<br>n = 108 | N/S | Individual, group or combination | N/S | N/S | **Job Satisfaction**<br>NSWQ<br>**Interview questions**<br>What can you tell me about your overall experience of CS? What, if any, benefits have you gained from CS? Has CS been of use to you in your practice and personal life? |

(*Continued*)

**Table 1.** (Continued)

| Study | Design | Setting | Participants | | Supervision | | | | Outcomes (Quantitative) OR Interview questions (Qualitative) |
|---|---|---|---|---|---|---|---|---|---|
| | | (country) | Supervisee **Profession** **Work Experience,** *mean* **n** | Supervisor **Profession** | Type | Frequency | Duration | | |
| McAuliffe 2013 | Quantitative Quasi-experimental cross sectional | Obstetric care settings (Africa) | Nursing/Medical N/S Cohort 1 n = 540 Cohort 2 n = 541 Cohort 3 n = 480 | N/S | N/S | N/S | N/S | | **Job Retention[a]** **Job Satisfaction** Job Satisfaction Scale |
| McCarron 2017 | Quantitative (Not included in the review) Qualitative Grounded theory | Psychiatric hospital (UK) | Nursing Cohort 1, 8.5 years n = 20 Cohort 2, 6.5 years n = 30 | N/S | N/S | N/S | N/S | | **No relevant outcomes** **Interview questions** What has your experience of CS been? If you feel that your level of CS is inadequate, how do you think this impacts on you, your ability to do your job and patient care? |
| Nathanson 1992 | Quantitative | Hospital and community services (USA) | Social work 50% ≤ 3 years n = 196 | Social work | Individual or group | N/S | N/S | | **Job Satisfaction[a]** |
| Saxby 2016 | Quantitative Cross sectional Qualitative Qualitative descriptive | Community health service (Australia) | Dietetics/Social Work/ Physiotherapy/ Podiatry/ Occupational Therapy/ Psychology/Speech Therapy 57% > 10 years n = 82 | N/S | Individual or group | N/S | N/S | | **Burnout** MBI **Job Retention** Intention to Leave Scale **Interview questions** How would you describe your experience of CS? What makes a CS effective? Any factors that reduce the effectiveness of CS? Can you give examples where CS has made a difference to: how services are delivered to clients? How workers cope with stresses in their job? how workers feel about where they work? |

(*Continued*)

**Table 1.** (Continued)

| Study | Design | Setting | Participants | | | Supervision | | | | Outcomes (Quantitative) OR Interview questions (Qualitative) |
|---|---|---|---|---|---|---|---|---|---|---|
| | | (country) | Supervisee | Supervisor | | Type | Frequency | Duration | | |
| | | | **Profession** | **Profession** | | | | | | |
| | | | **Work Experience,** *mean* | | | | | | | |
| | | | **n** | | | | | | | |
| Schroffel 1999 | Quantitative Cross sectional | Mental health service (USA) | Social Work/ Counselling/ Nursing/ Psychology | N/S | | Individual or group | Weekly | 71% > 30 minutes | | **Job Satisfaction** |
| | | | 16 years | | | | | | | JDI |
| | | | n = 84 | | | | | | | JIG |
| Severinsson 1996 | Quantitative Cross sectional | Psychiatric hospital (Sweden) | Nursing | Nursing | | Group | Weekly | 90 minutes | | **Work Environment**[a] |
| | | | 10 years | | | | | | | |
| | | | n = 26 | | | | | | | |
| Severinsson 1999 | Quantitative Quasi-experimental cross sectional | Acute hospital (Sweden) | Nursing | N/S | | N/S | N/S | N/S | | **Work Environment** |
| | | | N/S | | | | | | | Work Environment Measure |
| | | | n = 158 | | | | | | | |
| Teasdale 2001 | Quantitative Quasi-experimental cross sectional | Acute hospital and community health settings (UK) | Nursing | N/S | | Individual, group or combination | N/S | N/S | | **Burnout** |
| | | | 14 years | | | | | | | MBI |
| | | | n = 211 | | | | | | | **Work Environment** |
| | | | | | | | | | | Nursing in Context Questionnaire |
| Wallbank 2010 | Quantitative Randomised controlled trial | Acute hospital obstetrics and gynaecology (UK) | Nursing/Medical | Psychology | | Individual | N/S | 60 minutes | | **Burnout** |
| | | | N/S | | | | | | | ProQol |
| | | | n = 30 | | | | | | | **Well-being** |
| | | | | | | | | | | IES |
| | | | | | | | | | | ProQol |
| Webster 1999 | Quantitative Cross sectional | Community mental health services (USA) | Social Work/ Medical/ Psychology/ Counselling/ Nursing | N/S | | N/S | N/S | N/S | | **Burnout** |
| | | | N/S | | | | | | | MBI |
| | | | n = 151 | | | | | | | |
| White 1998 | Qualitative Qualitative descriptive | Community, medical ward, paediatric ward, management, School of Nursing, A&E department, gynaecology ward, GP unit, residential care (UK) | Nursing | Nursing | | Individual or group | N/S | N/S | | **Interview** |
| | | | N/S | | | | | | | **Questions** |
| | | | N = 12 | | | | | | | N/S |

a–outcome measure not validated; AJDI–Abridged Job Descriptive Index; BCS–Bradford Clinical Supervision Scale; CCQ–Creative Climate Questionnaire; CPQ–Copenhagen Psychosocial Questionnaire; GHQ–General Health Questionnaire; IES–Impact of Event Scale; IJSS–Intrinsic Job Satisfaction Scale; JDI–Job Descriptive Index; JIG–Job in General Index; MBI–Maslach Burnout Inventory; MBI-GS–Maslach Burnout Inventory-General Survey; SNCW–Satisfaction with Nursing Care and Work; NSWQ–Nursing Workplace Satisfaction Questionnaire; SF-36–36-Item Short Form Survey; ProQol–Professional Quality of Life Scale; QPSNordic–The Nordic Questionnaire for Psychological and Social Factors at Work; SOC–Sense of Coherence Scale; WEQ–Work Environment Questionnaire; WRSI–Work-related Strain Scale.

N/S–Not stated.

## Methodological quality

The predominant methodological risk of bias for analytical cross-sectional cohort studies (n = 14) was the absence of strategies to deal with confounding factors [36, 39, 41, 45, 46, 48, 52]. For quasi-experimental studies (n = 14) it was unclear if participants received similar support interventions other than clinical supervision in 12 studies [23–29, 31, 33–35, 51], outcome measurement was not performed both pre and post intervention (i.e. multiple time points) in nine studies [24, 27–29, 31, 33–35, 51], and it was unclear if participants were similar at baseline in seven studies [24, 27–29, 33, 34, 51]. The single randomised controlled trial [22] only met five of the 13 items; notably the method of randomisation was unclear and there was no between group statistical comparison. JBI Critical Appraisal Checklists can be found in S1–S3 Tables.

The methodological quality of the five qualitative studies (including the qualitative component of mixed methods studies) was generally high. Two studies [51, 52] scored 10 out of 10, while two other studies [49, 50] scored eight out of 10, failing to account for the potential influence of the researcher on the research findings. One study [53] did not demonstrate congruity between their stated philosophical perspective and the research methodology used, nor was there congruence between their research methodology and their research question/objectives, methods of data collection and analysis and interpretation of results. The JBI Critical Appraisal Checklist can be found in S4 Table.

## Impact of clinical supervision on organisational outcomes (quantitative findings)

**1. Clinical supervision compared to control.**   Eleven studies, including 2,965 participants, evaluated the effect of clinical supervision on organisational outcomes by comparison to a control group that did not receive clinical supervision [22, 24, 25, 27–29, 31, 33–35, 51]. Eight studies included nursing professionals [22, 24, 25, 31, 33–35, 51], one study included social work/psychology/counselling professionals [27], two studies included other allied health professionals [28, 29] and two studies included medical professionals [22, 33]. While individual studies found clinical supervision had a positive effect on organisational outcomes, there were variable results across studies for burnout (six studies, n = 776 participants) (Fig 2A–2D), job satisfaction (four studies, n = 2,020 participants), well-being (four studies, n = 444 participants), and workplace environment (five studies, n = 783 participants). Notably, a single randomised controlled trial (n = 30 participants) found that clinical supervision had a large effect on burnout (Fig 2D) and well-being [22]. Results from individual studies can be found in S5 Table.

**2. Clinical supervision compared to within-group historical control (pre/post implementation).**   Six studies, including 178 participants, evaluated the effect of clinical supervision on organisational outcomes by comparing post-implementation with pre-implementation [22, 23, 25, 26, 30, 32]. Six studies included nursing professionals [22, 23, 25, 26, 30, 32], one study included social work/psychology/counselling professionals [32] and one study included medical professionals [22]. While individual studies found clinical supervision had a positive effect on organisational outcomes, there were variable results across studies for burnout (four studies, n = 122 participants) (Fig 3A–3D), job satisfaction (four studies, n = 114 participants), well-being (five studies, n = 144 participants), and workplace environment (three studies, n = 95 participants). Results from individual studies can be found in S6 Table.

**3. Association between effective clinical supervision and organisational outcomes.** Eight studies, including 1,376 participants, investigated the association between effective clinical supervision and organisational outcomes [37, 38–43, 52]. Five studies included nursing

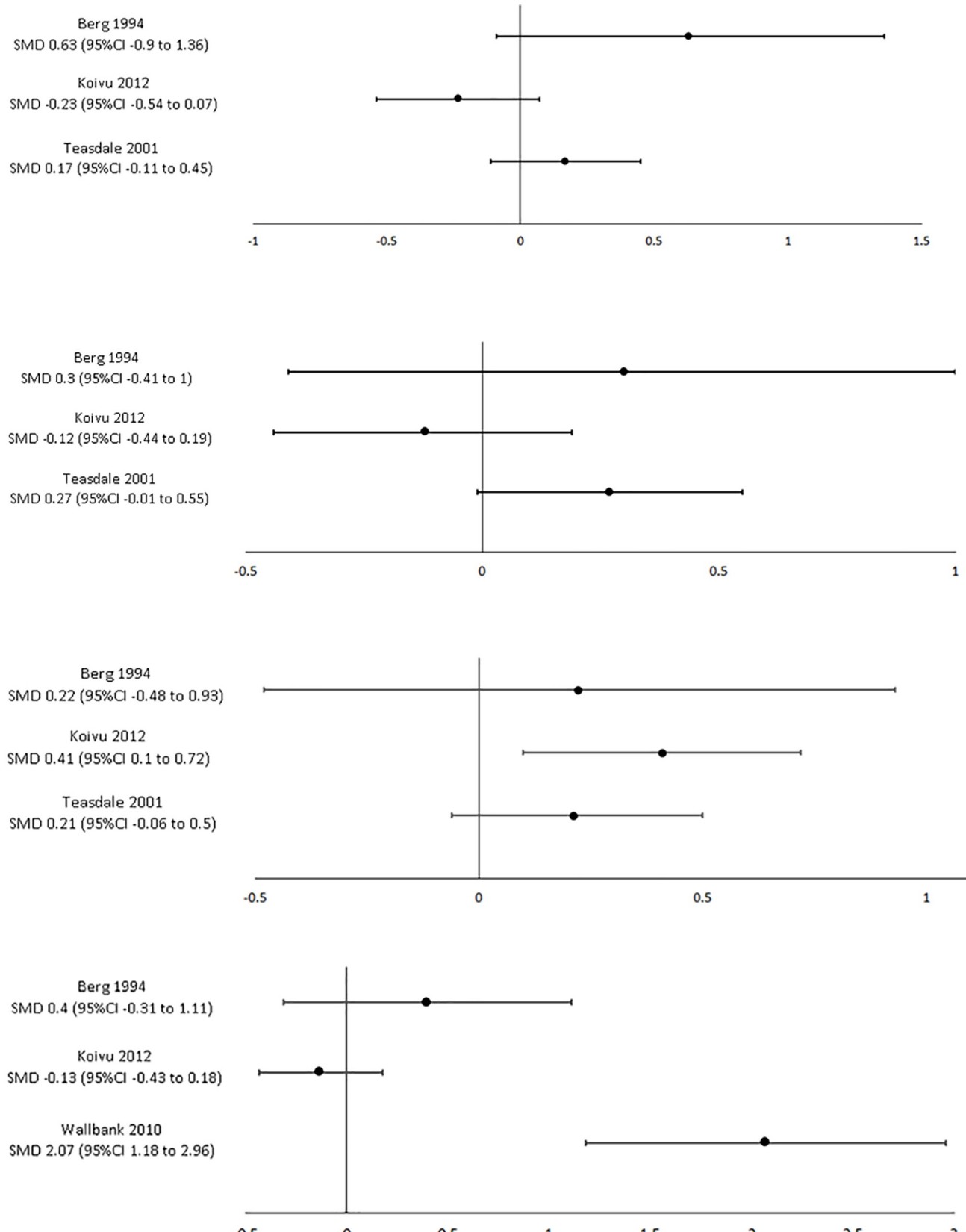

**Fig 2.** A. Supervision vs. control: emotional exhaustion (burnout) SMD 95%CI. B: Supervision vs. control: depersonalisation (burnout) SMD 95%CI. C: Supervision vs. control: personal accomplishment (burnout) SMD 95%CI. D: Supervision vs. control: overall burnout SMD 95%CI.

professionals [37, 39–42], three studies included social work/psychology/counselling professionals [38, 43, 52] and one study included other allied health professions [52]. There was preliminary evidence to suggest that effectiveness of clinical supervision may be negatively associated with burnout and positively associated with job retention (Table 2). The association between effective clinical supervision and job satisfaction was unclear.

Synthesis of five studies [37, 39, 41, 42, 52], including 1,046 participants, indicated that effectiveness of clinical supervision may be negatively associated with emotional exhaustion and depersonalisation, but not associated with personal accomplishment. Three studies found a small to moderate association with emotional exhaustion [39, 42, 52] and four studies found small association with depersonalisation [37, 39, 41, 42].

Synthesis of two studies [40, 52], including 152 participants, indicated that effectiveness of clinical supervision may be positively associated with job retention. Both studies found a moderate association with job retention.

Synthesis of three studies [41–43], including 836 participants, indicated that the association between effectiveness of clinical supervision and job satisfaction was unclear. Two studies [41, 42] found a small positive association and one study [43] found a small negative association. Results from individual studies are available in S7 Table.

**4. Association between effective supervisor and organisational outcomes.**   Eight studies, including 1,600 participants, investigated the association between effectiveness of the supervisor and organisational outcomes [32, 36, 39, 41, 43, 45, 46, 48]. Five studies included nursing professionals [32, 39, 41, 46, 48], seven studies included social work/psychology/counselling professionals [32, 36, 43, 45, 46, 48, 52] and one study included medical professionals [48]. There was preliminary evidence to suggest that an effective supervisor may be negatively associated with burnout, and positively associated with job satisfaction (Table 3).

Synthesis of three studies, [39, 41, 48] including 901 participants, indicated that an effective supervisor may be negatively associated with depersonalisation but not associated with personal accomplishment. Two studies found a small association with depersonalisation [39, 48]. The association between an effective supervisor and emotional exhaustion was unclear, with two studies finding a small negative association [39, 48] and one study finding a small positive association [41].

Synthesis of five studies [32, 41, 43, 45, 46], including 1128 participants, indicated that an effective supervisor may be positively associated with job satisfaction. Studies found a small to large association with job satisfaction. Results from individual studies are available in S8 Table.

## Healthcare professionals' experiences of clinical supervision as it relates to organizational processes and outcomes (qualitative findings)

Five studies, including two qualitative [49, 50] and three mixed methods studies [51–53], explored the experiences of healthcare professional supervisees on clinical supervision as it relates to organisational outcomes. A total of 16 findings and their illustrations were extracted. Of the 16 findings, 14 were unequivocal and two were credible. The 16 findings were organised into four categories which were further deduced to two synthesised findings. Table 4 shows a summary of the qualitative findings.

**Synthesised finding 1: Adequate clinical supervision mitigates the risk of burnout and facilitates staff retention, while inadequate clinical supervision can lead to stress and burnout.**   Health professionals indicated that if clinical supervision was adequate or if they felt

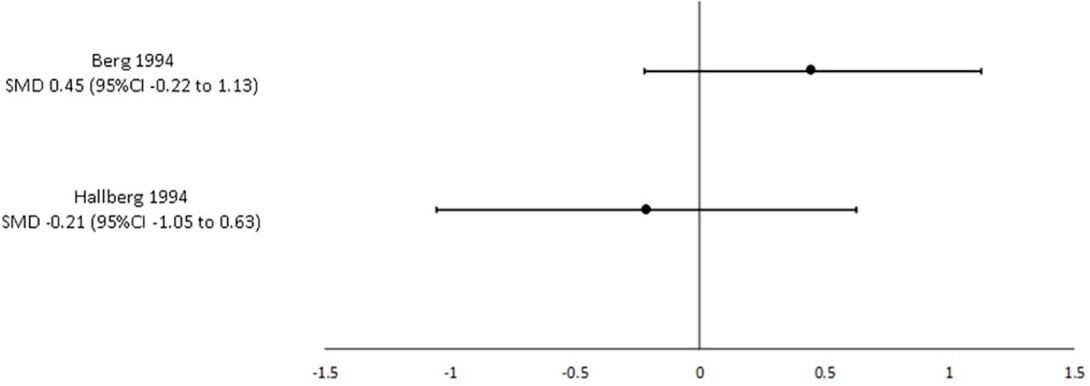

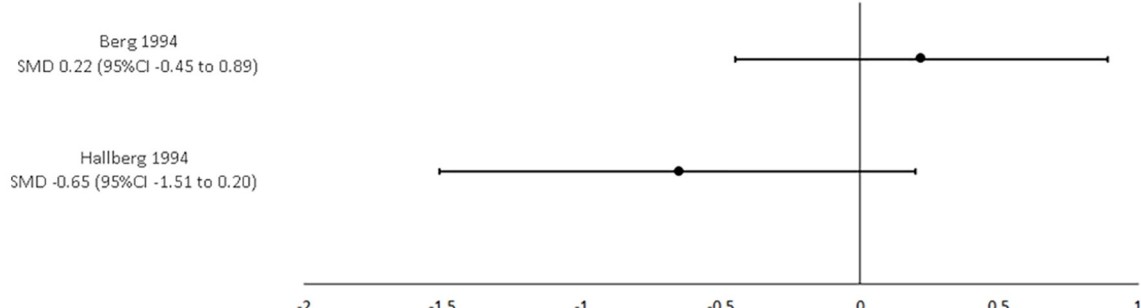

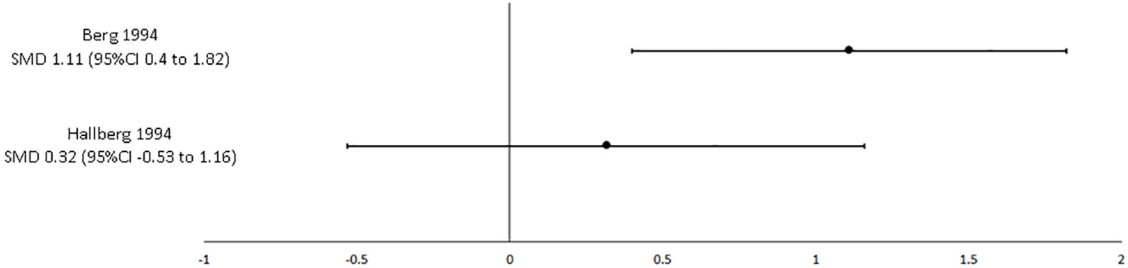

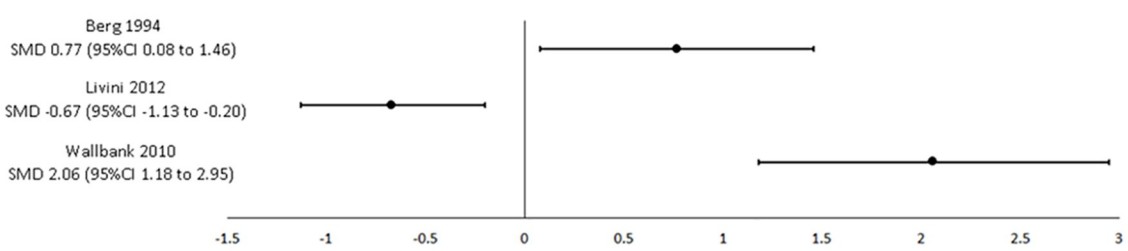

**Fig 3.** A: Pre- vs. post-supervision implementation: emotional exhaustion (burnout) SMD 95%CI. B: Pre- vs. post-supervision implementation: depersonalisation (burnout) SMD 95%CI. C: Pre- vs. post-supervision implementation: personal accomplishment (burnout) SMD 95%CI. D: Pre- vs. post-supervision implementation: overall burnout SMD 95%CI.

supported the risk of experiencing burnout or leaving the workplace was less likely. Conversely, health professionals, who felt that their supervision was inadequate, reported that clinical supervision had no positive impact or can lead to stress and burnout if they felt unsupported. This synthesised finding was developed from two categories comprising of seven unequivocal findings.

○ *Category 1.1 Adequate clinical supervision mitigates the risk of burnout, while inadequate clinical supervision can lead to stress and burnout.*

Adequate supervision meant that health professionals experienced the opportunity to debrief challenging events with their supervisor and gain a better understanding of patient interactions which can be stressful, and cause burnout for some staff. However, participants who felt unsupported identified stress and burnout as the negative consequences. This category was supported by four findings:

1. Some respondents felt that inadequate supervision had no impact; however, others identified personal consequences in terms of stress and burnout, feeling unsupported and there being an impact on their work, the ward, and clients. (Unequivocal)

2. Supervision assisted them to manage the workplace stress and hence, reduce their risk of burnout. (Unequivocal)

3. Supervision was helpful for the worker to gain a greater understanding of the dynamics operating in the client interaction to ensure there were no negative impacts for the worker or the client. (Unequivocal)

4. Opportunity to debrief challenging events provided supervisees with validation of their feelings and consideration of different management strategies to reduce their distress. (Unequivocal)

○ *Category 1.2 Implementation of effective clinical supervision facilitates staff retention and reduces their intention to leave.*

**Table 2. Synthesis of studies investigating association between effectiveness of clinical supervision and organisational outcomes.**

| Outcome | Number of studies | Number of participants | Direction of association within study (number of studies) | | | Effect size |
|---|---|---|---|---|---|---|
| | | | - | o | + | |
| Burnout–Emotional Exhaustion | 5 [37, 39, 41, 42, 52] | 1,046 | 3 | 2 | 1 | Small to moderate |
| Burnout–Depersonalisation | 5 [37, 39, 41, 42, 52] | 1,046 | 4 | 1 | 0 | Small |
| Burnout–Personal Accomplishment | 5 [37, 39, 41, 42, 52] | 1,046 | 1 | 3 | 1 | Moderate |
| Job Retention | 2 [40, 52] | 152 | 0 | 0 | 2 | Moderate |
| Job Satisfaction | 3 [41–43] | 836 | 1 | 0 | 2 | Small |
| Well-being | 1 [42] | 136 | 0 | 0 | 1 | U/A |

N/A–not applicable; U/A–Unable to calculate.

Positive association for job retention, job satisfaction, and well-being indicates effectiveness of supervision is associated with better outcome.

Negative association for burnout indicates effectiveness of supervision is associated with better outcome.

**Table 3. Synthesis of results: Association between an effective supervisor and organisational outcomes.**

| Outcome | Number of studies | Number of participants | Direction of association within study (number of studies) | | | Effect size |
|---|---|---|---|---|---|---|
| | | | - | o | + | |
| **Effectiveness of Supervisor** | | | | | | |
| Burnout–Emotional Exhaustion | 3 [39, 41, 48] | 901 | 2 | 0 | 1 | Small |
| Burnout–Depersonalisation | 3 [39, 41, 48] | 901 | 2 | 1 | 1 | Small |
| Burnout–Personal Accomplishment | 3 [39, 41, 48] | 901 | 0 | 3 | 0 | U/A |
| Burnout–Overall | 2 [32, 36] | 150 | 1 | 1 | 0 | Large |
| Job Satisfaction | 5 [32, 41, 43, 45, 46] | 1128 | 0 | 0 | 5 | Small to Large |
| Well-being | 2 [32, 36] | 180 | 0 | 1 | 1 | Large |

U/A–Unable to calculate.

Positive association for job satisfaction, role competence and well-being indicates effectiveness of supervision is associated with better outcome.

Negative association for burnout indicates effectiveness of supervision is associated with better outcome.

**Table 4. Summary of qualitative findings.**

| Synthesised Findings | Categories | Findings | Illustrations |
|---|---|---|---|
| *Synthesised Finding 1* | *Category 1.1* | Some respondents felt that inadequate supervision had no impact; however, others identified personal consequences in terms of stress and burnout, feeling unsupported and there being an impact on their work, the ward and clients. (UNEQUIVOCAL) | 'I feel my confidence is affected.' (RN 2016) (McCarron et al 2017, p. 153) |
| **Adequate clinical supervision mitigates the risk of burn out and facilitates staff retention, while inadequate clinical supervision can lead to stress and burnout.** | **Adequate CS mitigates the risk of burn-out, while inadequate CS can lead to stress and burnout** | Supervision assisted them to manage the workplace stress and hence, reduce their risk of burnout. (UNEQUIVOCAL) | 'When I first started with my supervisor I was in a really bad place. . . . and I was sort of at the point of no return, so getting my clinical supervision organized and constantly every month, that gave me back my confidence.' (Saxby 2016, p. 175) |
| | | Supervision was helpful for the worker to gain a greater understanding of the dynamics operating in the client interaction to ensure there were no negative impacts for the worker or the client. (UNEQUIVOCAL) | 'We're exploring. . . . the impact of that particular case on myself as a worker. . . .. it seems to make it clearer and give me insight into different ways of looking at that particular person.' (Saxby 2016, p. 175) |
| | | Opportunity to debrief challenging events provided supervisees with validation of their feelings and consideration of different management strategies to reduce their distress. (UNEQUIVOCAL) | 'I was absolutely gob-smacked with this new reform that could be coming in and potentially what could happen to me in terms of where I'm going to be going or that type of thing, you know, it's quite unsettling. . .. . . but just having that opportunity to debrief and face my concerns has been helpful.' (Saxby 2016, p. 175–176) |
| | *Category 1.2* | The implementation of clinical supervision as evidence that the health service management 'cared about' her and her colleagues and valued and wished to retain their workers. (UNEQUIVOCAL) | 'Yeah, it's supportive and I guess it's an indication the organization does care about us enough to push that. . . .. and they want to keep their staff.' (Saxby 2016, p. 173) |
| | **Implementation of effective CS facilitates staff retention and reduces their intention to leave** | Supervisees' responses illustrated that supervision did enhance job satisfaction and reduce workers' intention to leave. (UNEQUIVOCAL) | 'Now I feel like I can still cope with what's going on and that to me was worth it because otherwise I would probably be packing shelves at Coles or something. So it's given me back my self worth, just from supervision.' (Saxby 2016, p. 174) |
| | | The supervisor played an active role in encouraging staff to undertake career developing activities. (UNEQUIVOCAL) | 'I guess, encouragement, being encouraged to do something, maybe something that you didn't think you were capable of. . . . Yes, my supervisor. . .she's suggested I become a supervisor, so I've done that and I'm going to start doing that. Yes, she makes suggestions like that from a professional development point of view.' (Saxby 2016, p. 173) |

*(Continued)*

**Table 4.** (Continued)

| Synthesised Findings | Categories | Findings | Illustrations |
|---|---|---|---|
| *Synthesised Finding 2* | *Category 2.1*<br><br>**CS enhances team relationships through improved communication** | Midwives felt the structure of a safe space for regular reflection offered them continual opportunities for self-development especially in terms of enhanced communication and improved working relationships. (UNEQUIVOCAL) | *'For me personally it has helped with dealing with conflict stuff, and people, or my own personal issues with other people without ever having to involve them, because it was me that was able to adjust things.' (Midwife 6)* (Love et al 2017, p.278) |
| **CS improves the work environment through boosting staff morale, motivation to work, staff well-being and team relationships** |  | Midwives used words such as 'courage', 'confidence' and 'strength' to describe how their CS sessions had fostered in them an improved ability to engage in difficult conversations at work. (UNEQUIVOCAL) | *'I would have just been left in limbo with that situation and that person I think. So it enabled me to actually look at the situation and address it with that person.' (Midwife 2)* (Love et al 2017, p.278) |
|  | *Category 2.2* | Midwives described feeling more positive about the work environment with an increased desire to 'give back' to the unit. (CREDIBLE) | *'It really boosted morale and got people motivated.' (Midwife 3)* (Love et al 2017, p.278) |
|  | **CS promotes staff morale, motivation to work and well-being** | Prominent valuable outcomes of clinical supervision at the level of organization were the strengthened relationships with work colleagues, which on occasion was reported as a challenge for senior staff, and increased staff morale. (CREDIBLE) | *'I think if you affect staff morale, that in turn has got to affect patient morale, because the staff has such a strong influence over the patients. . . If the staff member feels supported, feels as if they've got somewhere to go, feel that they are not on their own and not isolated, which is how I think people do feel perhaps without the [clinical supervision] session, then you can sometimes unwittingly take it out on patients, I think. So I think it definitely affects patient care.' (Staff Nurse)* (White et al 1998, p. 190) |
|  |  | Enthusiasm, growth and organisational commitment were identified by supervisors and supervisees. (UNEQUIVOCAL) | *'. . . we did an evaluation just when we had our face to face meeting, she said that she's more enthusiastic about her position, she's more motivated, she's more organised and she's been encouraged to do more skills development activities.'* (Ducat et al 2016, p. 32) |
|  |  | Supervision kept workers motivated, interested and engaged in their roles of delivering health care services. These features of supervision increased allied health workers' sense of connection to the employing organisation and decreased their intention to leave. (UNEQUIVOCAL) | *'It's made such a difference to me as a practitioner. It helps you stay really focused on why am I here and it helps you stay focused on the positives that you are getting all the time because they are easy to forget about.'* (Saxby 2016, p. 171) |
|  |  | Receiving positive feedback was particularly valuable for workers (at the time of data collection) as they were experiencing high uncertainty in many areas including changes to their roles and the focus of the service. Feedback from supervisors provided reassurance, as well as a sense of stability amid the evolving occupational landscape. (UNEQUIVOCAL) | *'It's quite a supportive relationship, so your skills and your experience are recognised and that's quite important in the current environment when everything else is being questioned and changed all the time.'* (Saxby 2016, p. 172) |
|  |  | Supervision increased staffs' sense of connection to the employing organisation, enabling supervisees to feel that they individually had a place within the organisation and therefore a sense of belonging to something greater than their immediate and often atomized local environment. (UNEQUIVOCAL) | *'What it does bring is a sense of being connected to the broader organisation. To feel connected, it's just to feel connected to, that somebody has a clue what I do, that somebody thinks it's ok, that it's not just me floating around here hoping like crazy, I'm doing something useful. . . . . like I'm out there and nobody knows where I am or what I'm doing and that total sense of no one having you back almost. . . . . That feeling for me, the word is connected, to something bigger.'* (Saxby 2016, p. 173) |
|  |  | Improved evidence-based practice, best practice, patient safety and clinical governance were identified by managers, supervisors and clinicians. (UNEQUIVOCAL) | *'. . . and, we really do need to ensure that our clinicians are doing the best practice, that they are supported to develop the skills they need for the role they do, and to have someone to support them to do that, not just measure them against it. . .' (Ducat et al 2016, p.32)* |

Participants reported that clinical supervision was a reflection that the health organisation valued their staff. Participants also indicated that supervisors encouraged staff to pursue career developments. These experiences enhanced job satisfaction and reduced staffs' intention to leave the healthcare organisation. This category was supported by three findings:

1. The implementation of clinical supervision as evidence that the health service management 'cared about' her and her colleagues and valued and wished to retain their workers. (Unequivocal)

2. Supervisees' responses illustrated that supervision did enhance job satisfaction and reduce workers' intention to leave. (Unequivocal)

3. The supervisor played an active role in encouraging staff to undertake career developing activities. (Unequivocal)

**Synthesised finding 2: Clinical supervision improves the work environment through boosting of staff morale, motivation to work, staff well-being and team relationships.** Health professionals indicated that clinical supervision was valuable, led to increased motivation and enthusiasm at work, and provided not only reassurance to staff but also a safe space for improved working relationships. This synthesised finding was developed from two categories comprising of seven unequivocal findings and two credible findings.

○ *Category 2.1 Clinical supervision enhances team relationships through improved communication.*

Participants (ie. midwives) felt that clinical supervision offered an opportunity to enhance their ability to engage in difficult conversations with their team which is key in effective working relationships. This category was supported by two findings:

1. Midwives felt the structure of a safe space for regular reflection offered them continual opportunities for self-development especially in terms of enhanced communication and improved working relationships. (Unequivocal)

2. Midwives used words such as 'courage', 'confidence' and 'strength' to describe how their clinical supervision sessions had fostered in them an improved ability to engage in difficult conversations at work. (Unequivocal)

○ *Category 2.2 Clinical supervision promotes staff morale, motivation to work and well-being.*

Participants reported that having a clinical supervisor to support them and provide valuable feedback made them believe that they had a place within their organisation, increased their morale and enthusiasm at work, and improved their overall perception of their work environment. This category was supported by six findings:

1. Midwives described feeling more positive about the work environment with an increased desire to 'give back' to the unit. (Credible)

2. Prominent valuable outcomes of clinical supervision at the level of organization were the strengthened relationships with work colleagues, which on occasion was reported as a challenge for senior staff, and increased staff morale. (Credible)

3. Enthusiasm, growth and organisational commitment were identified by supervisors and supervisees. (Unequivocal)

4. Supervision kept workers motivated, interested, and engaged in their roles of delivering healthcare services. These features of supervision increased allied health workers' sense of connection to the employing organisation and decreased their intention to leave. (Unequivocal)

5. Receiving positive feedback was particularly valuable for workers (at the time of data collection) as they were experiencing high uncertainty in many areas including changes to their roles and the focus of the service. Feedback from supervisors provided reassurance, as well as a sense of stability amidst the evolving occupational landscape. (Unequivocal)

6. Supervision increased health professionals' sense of connection to the employing organisation, enabling supervisees to feel that they individually had a place within the organisation and therefore a sense of belonging to something greater than their immediate and often atomized local environment. (Unequivocal).

## Integration of quantitative and qualitative evidence

Quantitative and qualitative findings in this review have been largely complementary and supportive of each other, especially on the impact of clinical supervision on burnout, staff well-being, job satisfaction, job retention and workplace environment.

**Burnout.** Quantitative findings have provided preliminary evidence that effective clinical supervision and effective supervisor may be negatively associated with burnout. This was also supported by qualitative findings that showed that adequate clinical supervision mitigated the risk of burnout, and that inadequate clinical supervision lead to stress and burnout.

**Staff well-being.** Quantitative findings from a single randomised controlled trial showed a large effect on reducing burnout and enhancing well-being. Qualitative studies supported this, showing that effective clinical supervision improved staff well-being.

**Job satisfaction.** Although quantitative evidence from three studies showed that the association between effective clinical supervision and job satisfaction was unclear, evidence from four studies showed a positive association of an effective supervisor with job satisfaction. Qualitative findings supported this showing that effective clinical supervision strengthened team relationships and sense of belonging to the organisation, thereby enhancing job satisfaction. This was particularly true when the supervisor was effective, provided valuable feedback and encouraged staff to pursue career developments.

**Job retention.** Evidence from two quantitative studies showed a moderate positive association of the effectiveness of clinical supervision with job retention. Similarly, qualitative studies showed that adequate clinical supervision facilitated staff retention.

**Workplace environment.** Synthesis of quantitative evidence from 11 studies investigating the effect of clinical supervision, and six studies investigating post-implementation of clinical supervision with pre-implementation, showed variable results in regard to its effect on workplace environment. However, qualitative evidence highlighted that effective feedback from supervisors were considered valuable and improved supervisee perceptions of the work environment and their sense of belonging to the organisation.

In summary, both the quantitative and qualitative evidence highlight that effective clinical supervision and effective clinical supervisors may be associated with positive organisational outcomes, whereas, ineffective or inadequate clinical supervision and ineffective supervisors may have a negative impact on the well-being of the supervisee.

## Discussion

This systematic review of 32 studies is the first known synthesis of quantitative and qualitative evidence to further our knowledge on the impact from, and experiences of, clinical supervision of post-qualification health professionals, on organisational outcomes. Quantitative findings indicate that clinical supervision can have variable effects on organisational outcomes. The

effectiveness of both the clinical supervision and the supervisor appear to influence this effect; effective clinical supervision is associated with lower burnout and greater staff retention, and an effective supervisor is associated with lower burnout and greater job satisfaction. This is supported by the qualitative findings which show that healthcare professionals believe adequate clinical supervision can mitigate the risk of burnout, facilitate staff retention, and improve the work environment, while inadequate clinical supervision can lead to stress and burnout. Overall, qualitative synthesis highlights that the effectiveness of clinical supervision and supervisors can significantly influence the effect of clinical supervision on organisational outcomes.

Effective clinical supervision and effective supervisors may be pre-cursors for the realisation of beneficial effects of clinical supervision by healthcare organisations. This is consistent with a model of clinical supervision, for post-qualification health professionals, proposed by Gonge and Buss [42], where participation in effective clinical supervision (ie. prioritising supervision time) is a pre-requisite to beneficial clinical supervision. While clinical supervision has become increasingly mandated in many healthcare organisations, through standard policies and procedures, the subsequent challenge lies in its effective and consistent implementation and uptake. This can be achieved in several ways. Organisations can adopt/utilise evidence-informed clinical supervision frameworks to guide supervision, such as the one recently developed by Rothwell and colleagues [54]. This review by Rothwell and colleagues, based on evidence from 135 studies, encourages organisations to consider making supervision mandatory to increase the value placed on it, and provide protected time for supervisors and supervisees to engage with it. It also offers several practical strategies such as providing staff with both one-to-one and group supervision options, facilitating a person-centred supervision approach with clear boundaries, tasks, ground rules and record keeping processes, and provision of ongoing training to supervisors and supervisees [54]. Implementation and uptake of clinical supervision can be completed by building a positive organisational culture that supports engagement in and uptake of clinical supervision [54], which could be regularly monitored through routine evaluations. Such evaluations will be critical to identify and respond to what clinical supervision strategies have worked, or not worked, for whom and why. Based on our work in this field, we believe that the organisational context can have an important role, and there is no one-size fits all approach when it comes to supporting the implementation and uptake of clinical supervision within organisations.

Healthcare organisations also need to support clinical supervisors to build and foster positive supervisory relationships with their supervisees. This has commonly been reported to be the single most important factor that influences the effectiveness of clinical supervision [3, 11, 54], and requires investment of both time and resources. Supervisors and supervisees can also be guided by evidence-informed principles that facilitate effective clinical supervision. For example, Martin and colleagues [11] provide several practical recommendations for supervisors and supervisees to enhance the effectiveness of clinical supervision, such as the development of a supervision contract, undertaking sessions at an optimal length and frequency, utilising different modes including telesupervision, evaluating supervision, and working on skills and abilities such as open communication, flexibility, trust and availability to foster a positive supervisory relationship [11]. Health professionals can be provided with continuing professional development opportunities to upskill in evidence-informed supervision practices [3, 55]. There is evidence from a longitudinal, multi-methods study to support the delivery of supervision training in various modes such as videoconference, online and blended modes, thereby catering to those that can't access face-to-face training. In this study, participants knowledge and confidence in the provision of supervision increased after training, which was also sustained at three-months post-training across all the four modes. This success was

attributed to the careful design and delivery of training across different modes, which maximised participant access to training [56].

This review found various methodological concerns across many studies reviewed, which is consistent with findings from a recent survey of 20 systematic reviews on clinical supervision reported between 1995 to 2019 [3]. Methodological concerns include predominance of ex post facto, cross-sectional, correlational designs, small sample sizes, over reliance on self-report measures, lack of psychometrically sound supervision measures, and lack of experimental and longitudinal designs [3]. Incomplete provision of information (on clinical supervision parameters) seems to continue to plague supervision research, as again found in the survey of supervision reviews [3], and in the systematic review reported here, making it hard to judge the full merit of the study or replicate it. There is a need for further rigorous high-quality studies in this area that use pluralistic research approaches where experimental investigation, randomisation, and data-driven case studies are used in conjunction with ex post facto, and cross-sectional designs [3]. Studies also need to better define the specifics of the clinical supervision *intervention* to allow replication and identification of the clinical supervision practices that are, or are not, effective for improving outcomes.

## Limitations

The final review deviated from the protocol to also include group supervision, as many studies did not specify the type of supervision investigated. However, group supervision is commonly practiced in healthcare organisations and including these studies in this review likely improves the generalisability of our findings. Although the qualitative studies included were deemed to be of good quality, there were several shortcomings in the methodologies employed by the quantitative studies, especially the lack of randomised trials and absence of strategies to deal with confounding factors in cross-sectional studies. Although there were a variety of healthcare settings and health professionals represented in this review, the majority of included studies were conducted in mental health settings with nursing and/or mental health disciplines (i.e. psychology, counselling, and social work). This may limit the generalisability of the results to other disciplines and indicates the need for further research beyond mental health settings and nursing/mental health disciplines.

## Conclusions

Clinical supervision can have a variable effect on healthcare organisational outcomes. This effect appears to be influenced by the effectiveness of both the clinical supervision provided and that of the clinical supervisor. This highlights the need for organisations to invest in high quality supervision practices if they wish to benefit from clinical supervision. Without such investment, there is a risk of policy-practice gaps in this area (i.e. while there may be policies to support clinical supervision in healthcare organisations, in practice it may not be implemented well). Ongoing further research, which grows the evidence base for high quality clinical supervision and helps to unpack the *black box* of clinical supervision practices that have the most effect on organisational outcomes, is required.

## Supporting information

**S1 Checklist. PRISMA checklist.**
(DOC)

**S1 Table. JBI critical appraisal checklist for randomised controlled trials.**
(DOCX)

**S2 Table. JBI critical appraisal checklist for quasi-experimental studies.**
(DOCX)

**S3 Table. JBI critical appraisal checklist for analytical cross sectional studies.**
(DOCX)

**S4 Table. JBI critical appraisal checklist for qualitative studies (including qualitative component of mixed methods studies).**
(DOCX)

**S5 Table. Results of studies investigating the effect of clinical supervision on organisational outcomes compared to control (no supervision).**
(DOCX)

**S6 Table. Results of studies investigating the effect of clinical supervision on organisational outcomes pre/post implementation.**
(DOCX)

**S7 Table. Results of studies investigating the association between effectiveness of clinical supervision and organisational outcomes.**
(DOCX)

**S8 Table. Results of studies investigating the association between an effective supervisor and organisational outcomes.**
(DOCX)

**S1 Appendix. Search strategy.**
(DOCX)

**S2 Appendix. Excluded studies.**
(DOCX)

## Acknowledgments

The authors would like to thank Ms Esther Tian for assistance with the literature search.

## Author Contributions

**Conceptualization:** Priya Martin, Lucylynn Lizarondo, Saravana Kumar, David Snowdon.

**Data curation:** Priya Martin, Lucylynn Lizarondo, Saravana Kumar, David Snowdon.

**Formal analysis:** Priya Martin, Lucylynn Lizarondo, Saravana Kumar, David Snowdon.

**Investigation:** Priya Martin, Lucylynn Lizarondo, Saravana Kumar, David Snowdon.

**Methodology:** Priya Martin, Lucylynn Lizarondo, Saravana Kumar, David Snowdon.

**Project administration:** Priya Martin.

**Resources:** Priya Martin, Lucylynn Lizarondo, Saravana Kumar, David Snowdon.

**Validation:** Priya Martin, Lucylynn Lizarondo, Saravana Kumar, David Snowdon.

**Visualization:** David Snowdon.

**Writing – original draft:** Priya Martin, Lucylynn Lizarondo, Saravana Kumar, David Snowdon.

**Writing – review & editing:** Priya Martin, Lucylynn Lizarondo, Saravana Kumar, David Snowdon.

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
