## [Decision Letter · Decision Letter 0]

3 Aug 2021

PONE-D-21-20050

Impact of clinical supervision on healthcare organisational outcomes: a mixed methods systematic review

PLOS ONE

Dear Dr. Martin,

Thank you for submitting your manuscript to PLOS ONE. After careful consideration, we feel that it has merit but does not fully meet PLOS ONE’s publication criteria as it currently stands. Therefore, we invite you to submit a revised version of the manuscript that addresses the points raised during the review process.

Fortuitously for this manuscript I was able to elicit the input of four experienced content experts to provide their peer review of the submission. All four found merit with the submission and have offered detailed feedback, as below, to assist refining the message for the audience.

We look forward to receiving your revised manuscript.

Kind regards,

Shane Patman, PhD

Academic Editor

PLOS ONE

Reviewers' comments:

Reviewer's Responses to Questions

**Comments to the Author**

1. Is the manuscript technically sound, and do the data support the conclusions?

Reviewer #1: Yes

Reviewer #2: Yes

Reviewer #3: Yes

Reviewer #4: Yes

2. Has the statistical analysis been performed appropriately and rigorously? 

Reviewer #1: Yes

Reviewer #2: Yes

Reviewer #3: I Don't Know

Reviewer #4: N/A

3. Have the authors made all data underlying the findings in their manuscript fully available?

Reviewer #1: Yes

Reviewer #2: Yes

Reviewer #3: Yes

Reviewer #4: Yes

4. Is the manuscript presented in an intelligible fashion and written in standard English?

Reviewer #1: Yes

Reviewer #2: Yes

Reviewer #3: Yes

Reviewer #4: Yes

5. Review Comments to the Author

Reviewer #1: This comprehensive review explored an important topic. I enjoyed reading this manuscript which clearly synthesised the literature. I have some suggestions to enhance clarity. Thank you for synthesising and contributing to the evidence.

Abstract:

L80 supervisor should be supervision?

Intro:

L93-94 needs reference

L116-130 Rather than reporting each review, can you combine the outcomes and reference accordingly?

Methods:

On the whole the methods were clearly described. It would be beneficial to outline the philosophical framework (ie pragmatism?) for the meta-aggregative approach to help situate the reader at the outset. This reference provides excellent context: Hannes, K. and Lockwood, C., 2011. Pragmatism as the philosophical foundation for the Joanna Briggs meta‐aggregative approach to qualitative evidence synthesis. Journal of advanced nursing, 67(7), pp.1632-1642.

L217 why were 3 levels of credibility assigned, and how was this decided? Quality appraisal was already conducted so it is not clear why there is an additional layer to this.

L241 Reference 21 does not relate to the meta-aggregative approach. This is perhaps why the above point requires clarifying

L241-255 How many authors conducted the qualitative synthesis?

Results:

Quantitative findings were comprehensively reported

L385 How was it determined that findings were credible or unequivocal? This was not described in the methods.

L406-409 Perhaps it is my lack of understanding regarding the “unequivocal” status, but finding 1 seems to indicate that supervision may or may not have an impact. So is the unequivocal finding that there is variability in supervisees’ perceptions of impact?

Discussion:

The discussion reads well and considers important aspects to translate the findings of this review.

L532-534 This sentence is not quite clear – it seems to state the obvious, as how can one experience beneficial supervision without participating in supervision?

L539 Which review?

L526-527 Is it inadequate supervision that leads to stress/burnout? Or the workload itself?

L604-605 Not sure that “The direction of this effect appears to be influenced by the effectiveness of both the clinical supervision provided and that of the clinical supervisor” when these data were not extracted from the study.

A number of interesting and practical points were made. It seems clear that clinical supervision is beneficial but concepts of cost to organisations (or individuals) was not considered. What are the barriers to implementing effective clinical supervision programs when there is clearly much evidence to support their benefits?

Reviewer #2: I have thoroughly enjoyed reading this systematic review. The authors are to be commended for their comprehensive review. I have only minor comments to align the manuscript to the PRISMA 2020 expanded checklist: PRISMA_2020_expanded_checklist.pdf (prisma-statement.org)

Strengths of the manuscript:

Introduction

Clear and logical development of the research questions that underpinned the systematic review.

Methods section

Protocol registration – stated as reference 15 Martin et al (2020) JBI Evid Synth. 2020;18(1):115-120. doi: 10.11124/JBISRIR-D-19-00017. PMID:658 31464853

Eligibility criteria - listed as inclusion criteria. I would recommend changing the sub-title from inclusion criteria and using the term eligibility criteria as per both 2009 and 2020 PRISMA Checklists. Comprehensive table of excluded studies and the respective reason is presented in Appendix 2.

Information sources – presented in Appendix 1 and main body of the manuscript.

Data items - Great clarity is provided regarding extraction of the qual, quant and mixed methods data

Effect measures - the outcomes. Used in the synthesis and presentation of the results, including thresholds, are presented in the manuscript.

Synthesis of results - Data synthesis and integration are comprehensively presented.

Results section:

Study selection & study characteristics are both clearly articulated for the 32 included studies. Table 1 clearly presents the study characteristics. Table 2. Clearly presents the synthesis of studies investigating association between effectiveness of clinical supervision and organisational outcomes.

Results of individual studies – comprehensive review and presentation within the manuscript, tables, and supplementary tables. The authors are commended for such a comprehensive presentation of the qualitative and quantitative review.

Methodological quality results are presented on page 13. Supplementary table four presents the JBI critical analysis for qualitative studies and includes a qualitative component of mixed methods studies.

Results synthesis – Extensive provision of supplementary tables 1-8 illustrate the breadth and depth of data synthesis from the included studies, according to the associations of outcome measures.

Integration of qualitative and quantitative evidence is well constructed and logical.

Discussion section: The findings are interpreted in the context of other evidence, including limitations of the evidence and the review.

Support: No funding was utilised to undertake this systematic review.

Availability of data, code, and other materials: A comprehensive range of supporting documents have been provided.

Areas for consideration:

Search strategy – Comprehensive search strategy provided in Appendix 1. As this is a mixed methods review, it would be ideal to clarify if the search strategy was developed If the search strategy structure adopted a PICO-style approach/SPIDER/SPICER, then describe the final conceptual structure and any explorations that were undertaken to achieve it.

Study selection – reviewer roles during the screening have been specified. Please clarify additional PRISMA 2020 checklist (section 16) reporting recommendations:

• Report any processes used to obtain or confirm relevant information from study investigators.

• If abstracts or articles required translation into another language to determine their eligibility, report how these were translated

Study risk of bias assessment - Quality assessment sub-header is stated, and reviewer involved noted. All studies were reviewed regardless of their methodological quality. It is not clear why the Mixed Methods Assessment Tool (MMAT) was not utilised for example, for the included mixed methods studies. The JBI Qualitative component is presented in supplementary Table 4.

Data extraction sub-heading is used instead of data collection process (PRISMA criteria 10) and Data items (PRISMA Criteria 11). Consider revising the subheadings to align better with the PRISMA 2020 checklist. The role and number of reviewers involved are reported within the manuscript.

Certainty assessment is not overtly covered in the current manuscript (sections 15 and 22 of the PRISMA (2020) expanded checklist

Results: With respect to Supplementary Table 6 the Maslach burnout inventory scores are not individualised with respect to Livini (2012), is this an oversight or where they not separated in the research study. Perhaps worth clarifying.

There is an excellent summary of qualitative findings in Table 4. To enhance the comparison of information in the main body of the manuscript and Table, it would be helpful to indicate the categories: e.g., “Category 1.2 Implementation of effective clinical supervision facilitates staff retention and reduces their intention to leave…” stated on lines 418-419, but the table does not have numbered categories in column 2 of Table 4.

The implications of the results for practice and policy are not overtly presented. Please make explicit recommendations for future research, as per section 23d of the PRISMA (2020) expanded checklist.

Reviewer #3: This manuscript presents a much-needed systematic review of the quantitative, qualitative, and mixed methods supervision literature capturing organizational outcomes in healthcare. The authors have conducted a methodologically sound, comprehensive review which will contribute to the interdisciplinary healthcare and supervision literature, and support practice. Additional information and clarity as noted below would be beneficial.

Introduction

In justifying the need for the study, the authors nicely present the gap in the supervision literature related to reviews of organizational outcomes, noting reviews typically focus on client or practitioner outcomes. An additional comment/value statement related to why reviewing organizational supervision outcomes is important outside of it not being done would highlight the contribution of this manuscript.

Methods:

1. given the definition provided, one would assume “group supervision” was led be a supervisor, and therefore “peer-supervision” only studies were not included. Can this be explicitly noted?

2. clarify “date of inception” pg. 7, line 176;

3. include date range of studies reviewed;

4. a brief definition of “pearling” on page 8 would be helpful.

Results:

5. I very much appreciated the detail presented, however, at times the narrative description is difficult to follow. A statement regarding the presentation (e.g., number of studies in specific categories are not mutually exclusive) would be helpful.

6. The sentence regarding setting is confusing as it seems to be reporting on both setting and profession. I found myself trying to understand if they were connected. Separate sentences per characteristic would make the meaning more clear, unless they are specifically connected in which case outlining the connection would be helpful.

7. In presenting the type of supervision, the total number of studies presented is 30, leaving me curious about the other two studies. Additionally, I was not sure why multi-professional supervision was separated out related to this characteristic?

8. For frequency and duration of supervision, can the number of studies included in the mean be reported explicitly? Or conversely the number that did not?

9. The number of participants represented in the studies as connected to methodological quality is helpful. Including the number of participants included in all categories of impact would be useful (e.g., adding to control, and within-group sections on page 14 and 15).

10. What definitions were used for “effective supervision” and “effective supervisor”?

11. For the synthesized finding, Category 2.1: was the finding about impact on teams specific to group supervision? Can this be noted? Midwives are also specifically noted in this finding, yet other disciplines are not noted at any other point in the Results. This leads to questions related to specifics of this discipline, and homogeneity of findings across the others. Can this please be clarified?

12. In the study protocol, authors identify the range of professions to be covered in this review. Aside from the above results related to midwives, and summary of nurses, psychology, social workers, and counsellors, there is little comment on the professions representation in this literature. Addressing this would be important.

Reviewer #4: Thank you for the opportunity to review this interesting manuscript. Overall I found the work to be coherent and well constructed. My major comment would be clarification in the title and throughout the manuscript that the work focuses on the post-professional clinical supervision literature, rather than the supervision literature as a whole (including pre-professional students). The methods section could also be reduced as much of this is described in the protocol paper but focus on the elements that were modified from the protocol. I have also made other suggestions throughout in the attached document. This will be a useful addition to the literature and thanks for the contribution.

6. PLOS authors have the option to publish the peer review history of their article (what does this mean?). If published, this will include your full peer review and any attached files.

Reviewer #1: **Yes: **Simone Gibson

Reviewer #2: No

Reviewer #3: No

Reviewer #4: **Yes: **Brett Vaughan

---

## [Decision Letter · Decision Letter 1]

4 Nov 2021

Impact of clinical supervision on healthcare organisational outcomes: a mixed methods systematic review

PONE-D-21-20050R1

Dear Dr. Martin,

We’re pleased to inform you that your manuscript has been judged scientifically suitable for publication and will be formally accepted for publication once it meets all outstanding technical requirements.

Kind regards,

Shane Patman, PhD

Academic Editor

PLOS ONE

Additional Editor Comments (optional):

Reviewers' comments:

Reviewer's Responses to Questions

**Comments to the Author**

1. If the authors have adequately addressed your comments raised in a previous round of review and you feel that this manuscript is now acceptable for publication, you may indicate that here to bypass the “Comments to the Author” section, enter your conflict of interest statement in the “Confidential to Editor” section, and submit your "Accept" recommendation.

Reviewer #2: All comments have been addressed

Reviewer #3: (No Response)

Reviewer #4: All comments have been addressed

2. Is the manuscript technically sound, and do the data support the conclusions?

Reviewer #2: Yes

Reviewer #3: Yes

Reviewer #4: Yes

3. Has the statistical analysis been performed appropriately and rigorously? 

Reviewer #2: Yes

Reviewer #3: I Don't Know

Reviewer #4: N/A

4. Have the authors made all data underlying the findings in their manuscript fully available?

Reviewer #2: Yes

Reviewer #3: Yes

Reviewer #4: Yes

5. Is the manuscript presented in an intelligible fashion and written in standard English?

Reviewer #2: Yes

Reviewer #3: Yes

Reviewer #4: Yes

6. Review Comments to the Author

Reviewer #2: I would like to thank you for taking the time to make the adjustments to your manuscript and accompanying files. The reviewers comments have been responded to appropriately and comprehensive information provided if a change was not made.

Reviewer #3: The paper is well-written, the study rigourous, and will make a significant contribution to the literature. I have one outstanding comment related to my earlier review:

• Comment #10: I appreciate the explanation of the reliance on the definitions of effectiveness provided by the authors’ of the studies included in the review. I believe this is important to note in the actual manuscript.

Reviewer #4: Thank you for the opportunity to review the revised version of this manuscript. The authors have satisfactorily addressed the comments on the previous version. This manuscript will be a useful addition to the clinical supervision literature and I look forward to seeing it published.

7. PLOS authors have the option to publish the peer review history of their article (what does this mean?). If published, this will include your full peer review and any attached files.

Reviewer #2: No

Reviewer #3: No

Reviewer #4: **Yes: **Brett Vaughan

---

## [Editor Report · Acceptance letter]

9 Nov 2021

PONE-D-21-20050R1 

Impact of clinical supervision on healthcare organisational outcomes: a mixed methods systematic review 

Dear Dr. Martin:

I'm pleased to inform you that your manuscript has been deemed suitable for publication in PLOS ONE. Congratulations! Your manuscript is now with our production department. 

Kind regards, 

on behalf of

Assoc Prof Shane Patman 

Academic Editor

PLOS ONE